# Irritable Bowel Syndrome and the Gut Microbiome: A Comprehensive Review

**DOI:** 10.3390/jcm12072558

**Published:** 2023-03-28

**Authors:** Sofia D. Shaikh, Natalie Sun, Andrew Canakis, William Y. Park, Horst Christian Weber

**Affiliations:** 1Department of Medicine, Chobanian & Avedisian School of Medicine, Boston University, Boston, MA 02118, USA; 2Division of Gastroenterology and Hepatology, University of Maryland School of Medicine, Baltimore, MD 21201, USA; 3Division of Gastroenterology, The Warren Alpert Medical School, Brown University, Providence, RI 02903, USA; 4Section of Gastroenterology, Chobanian & Avedisian School of Medicine, Boston University, Boston, MA 02118, USA; 5Section of Gastroenterology, VA Boston Healthcare System, Boston, MA 02130, USA

**Keywords:** irritable bowel syndrome, gut microbiome, fecal microbiota transplantation, FODMAP diet, brain gut axis

## Abstract

Irritable Bowel Syndrome (IBS) is a functional disorder of the gastrointestinal tract characterized by abdominal pain and altered bowel habits. It has a prevalence of 10 to 25% in the United States and has a high disease burden, as evidenced by reduced quality of life, decreased work productivity and increased healthcare utilization and costs. IBS has been associated with several intra-intestinal and extra-intestinal conditions, including psychiatric comorbidities. Although the pathophysiology of IBS has not been fully elucidated, it involves dysregulation of communication between the brain and gut (brain–gut axis) which is associated with alterations in intestinal motility, gut permeability, visceral hypersensitivity and gut microbiota composition. The purpose of this article is to review the role the gut microbiota plays in the pathophysiology of IBS, understand factors that affect the gut microbiome and explore the microbiome as a target of treatment.

## 1. Introduction

Irritable Bowel Syndrome (IBS) is a functional gastrointestinal disorder (FGID) characterized by chronic abdominal pain or discomfort and altered bowel habits. It has an estimated prevalence of 10 to 25% of the United States, with significant geographic variability, with the highest rates in South America (17–21%) and the lowest rates in South Asia (7–9%) and 5.6% in the Middle East and Africa [1,2]. It is estimated that over 40% of people worldwide meet criteria for a FGID, including (but not limited to) IBS [3]. IBS disproportionately affects females compared to males: 1.5- to 3-fold [2,4]. It occurs in patients of all age groups, with onset of symptoms by the age of 35 in 50% of patients [2] and decreasing prevalence in individuals over the age of 50 [4].

IBS is classified as a FGID, meaning that its associated gastrointestinal symptoms cannot be attributed to a specific structural or metabolic abnormality. A diagnosis of IBS is clinically determined based on the Rome IV criteria, which defines IBS as recurrent abdominal pain that occurred, on average, at least one day per week in the last three months, involving two or more of the following associated symptoms: defecation, change in stool frequency or change in stool appearance [5]. IBS can be further broken down into sub-classes based on the Bristol Stool Form Scale (BSFS) which characterizes stool consistency from hard to soft on a scale of 1–7 [5,6,7]. The subclasses of IBS include IBS with predominant constipation (IBS-C), IBS with predominant diarrhea (IBS-D), IBS with mixed bowel habits (IBS-M) and unclassified IBS (IBS-U). IBS-C is typically seen with type 1–2 in BSFS and IBS-D with type 6–7 in BSFS. Patients with IBS m have manifestations of both constipation and diarrhea. IBS-U meets the diagnostic criteria for IBS but cannot be classified within the other three subtypes [5,6,7].

IBS has a high disease burden, accounting for at least 20% of outpatient gastroenterology clinic referrals [2]. Annually, there are more than 54.4 million ambulatory visits in the US for primary GI diagnoses [8]. Although limited data exist on the prevalence of IBS in primary care clinics, one study of 3111 patients seeing general practitioners in the UK found that of the 255 who presented with a primary GI complaints, 30% were deemed to have IBS and 14% had other functional bowel disorders [9]. IBS has been associated with increased healthcare costs and resource utilization [10]. Notably, annual healthcare related expenses have been found to be ~50% higher in patients with IBS, accounting for the cost of clinic visits, medications, radiology, as well as laboratory testing [10]. IBS has also been associated with increased risk of extra-abdominal and abdominal surgery, with patients being three times more likely to have a cholecystectomy and twice as likely to have an appendectomy or hysterectomy [11].

Increasing evidence has illustrated that IBS not only manifests with GI symptoms, but also significantly affects emotional health and wellbeing, with the majority of IBS patients experiencing extra-intestinal manifestations, including clinically significant psychiatric disease [2]. IBS has routinely been associated with higher levels of stress, reduced quality of life and decreased work productivity [2,12].

Despite not being well understood, the pathophysiology behind IBS is believed to be multifactorial based on genetic, dietary, gastrointestinal and central nervous system influences (Figure 1) [13,14]. Recent studies have examined factors such as alterations in the gut microbiota (dysbiosis), changes in gut motility and mucosal inflammation as well as the role of the central nervous system, including visceral hypersensitivity and the gut–brain axis (BGA) [13,14]. As the microbiome genomic database has expanded, so too has our understanding of the role of the microbiome in bidirectional signaling via the gut–brain axis. The gut microbiome may serve as a promising therapeutic target in the management of IBS in the future. The goal of this paper is to provide a comprehensive review of IBS with a focus on the emerging role of the microbiome on its pathogenesis and the potential therapeutic strategies for treating this condition.

Several factors have been implicated in the pathogenesis of IBS, including dysregulated communication between the brain and gut (brain–gut axis), changes in the gut microbiome, prior infections, psychological stress, diet, as well as pre- and postnatal factors.

## 2. Gastrointestinal and Extra-Intestinal Conditions Associated with IBS

### 2.1. Gastrointestinal Comorbidities of IBS

IBS frequently co-occurs with other gastrointestinal as well as extra-intestinal comorbidities. Although IBS has been linked to several other gastrointestinal conditions, the association has been best characterized between IBS and functional dyspepsia (FD) and gastroesophageal reflux disorder (GERD). A cross-sectional study of 1443 subjects found the prevalence of IBS, GERD and dyspepsia to be 9.6%, 8.5% and 9.5%, respectively [15]. Out of the patients who had IBS, 21% had GERD and 14% dyspepsia [15]. One meta-analysis found that the prevalence of IBS in patients with FD was 37% compared to 7% of patients without FD [16]. In the same sample, subjects with FD had an overall eight-fold increase in IBS compared to patients without [16]. Additional studies have found the odds ratio of concurrent diagnoses of IBS and GERD to be significant, ranging from 3 to 16, indicating a clear relationship between the two conditions [17,18]. There is also evidence that the co-occurrence between these conditions is greater in patients who have anxiety [15].

### 2.2. Extraintestinal Comorbidities IBS

IBS has also been associated with several extra-intestinal comorbidities, most notably chronic pain, chronic fatigue syndrome as well as several psychiatric conditions. The association between IBS and fibromyalgia, a chronic pain disorder described as widespread musculoskeletal pain in the absence of muscle or joint inflammation, has also been well characterized. The prevalence of fibromyalgia in IBS patients is generally believed to be 31.6 to 63%, with some estimates as high as 81% [19,20]. Based on the Functional Bowel Disorder Severity Index (FBDSI), a validated tool which can be used to evaluate disease severity among IBS patients, patients with concomitant IBS and fibromyalgia had significantly higher FBDSI scores compared to patients with IBS only [21]. There is also substantial evidence to support an association between IBS and other chronic pain disorders. Studies have found that 35–40% of patients with chronic pelvic pain meet the criteria for IBS [22,23]. IBS patients have been found to be three times as likely to have temporomandibular pain [24] and 25–50% of IBS patients report migraines or headaches [25].

In addition to chronic pain, chronic fatigue has also been consistently associated with IBS. Chronic fatigue syndrome, also known as myalgic encephalomyelitis, is a condition characterized by extreme fatigue that cannot be attributed to any underlying medical condition. The prevalence of chronic fatigue syndrome in the general population tends to be quite low, with self-reported prevalence of 3.28% [26] and prevalence of less than 1% based on clinical assessment [26,27]. The lifetime prevalence of IBS has been estimated to be 92% in patients with chronic fatigue syndrome [28] and the odds ratio of concurrent IBS with clinically diagnosed chronic fatigue syndrome is estimated to be 6.5. This is noted to be even higher in presumptive and self-reported cases of chronic fatigue [29].

### 2.3. Psychiatric Comorbidities of IBS

It is well accepted that there is an association between IBS and psychiatric comorbidities. Studies have found 44% [30] of IBS patients have a psychiatric diagnosis, while up to 69.6% [31] have a psychological comorbidity (i.e., high anxiety or depression scores). Psychological comorbidities in IBS patients are associated with poorer prognosis, and patients are also more likely to have seen a gastroenterologist and to have trialed more treatment options [31].

Among patients who meet criteria for IBS, 26–45.5% have symptoms consistent with depression and 30–39.1% have symptoms consistent with anxiety [30,32,33]. These rates are significantly higher than estimates of depression (6.4–21%) [33,34,35] and anxiety (7.3–20.8%) [33,36] in the general population. One study of 2005 participants found that generalized anxiety disorder (based on DSM-IV criteria) was more than five times more prevalent in patients with IBS (OR: 5.84, *p* < 0.001) [37].

Despite several studies linking IBS to mood and anxiety disorders, there are minimal data describing the association between PTSD and IBS. One cohort of 50 patients who met criteria for IBS found that 44% of patients had a history of trauma and 36% were diagnosed with PTSD [38]. Similarly, a study of 339 female veterans found that 51% of female veterans with symptoms consistent with IBS met the criteria for PTSD (based on the Mississippi Scale for Combat Related PTSD score) in comparison to 30.9% of veterans without IBS [39]. Another study of 337 female veterans receiving care at the VA revealed history of prior trauma to be an independent risk factor for development of IBS, with a 50–115% excess risk dependent on trauma type [40]. Specifically, there was a clear association between prior sexual trauma and IBS [40,41].

While it is clear that there is an association between IBS symptoms and psychological comorbidities, the directionality of the relationship is not known. It has been speculated that the reduced quality of life of people with IBS [42,43,44] may lead to the higher rates of depression and anxiety in this population. However, it is also likely that psychiatric condition may affect IBS symptoms directly via the BGA, as discussed below. During the COVID-19 pandemic, individuals with IBS have been found to have both worsening of psychological and gastrointestinal symptoms [43,45]. One study found that individuals with a diagnosis of IBS according to the Rome IV criteria reported increased levels of anxiety symptoms (81%) based on GAD-7 scores and depressive symptoms (67%) based on PHQ-9 scores [43]. They were also noted to have worsening abdominal pain (48%), diarrhea (45%) and constipation (44%) [43].

## 3. Pathophysiology of IBS

The pathophysiology of functional GI disorders, including IBS, is intricate and multifactorial. Although not well understood, it is believed to involve dysregulation of the autonomic nervous system, as well as gastrointestinal, environmental and genetic factors [13,46]. Interactions between the central nervous system and enteric nervous system form a feedback loop via the BGA, which has been implicated in the development of many of the changes in gut function that lead to IBS [46]. The enteric nervous system (ENS), which makes up the largest component of peripheral nervous system, is a neural network compromised of ganglionic and aganglionic plexuses embedded within the gut wall, which plays an integral part in gastrointestinal function. The ENS is made up of enteric neurons and glial cells that play a role in gut motility as well as secretory, absorptive and immune regulation. Dysregulation of the ENS has been associated with several digestive disorders, and more recently, it has been implicated in the development of non-gastrointestinal conditions, including neurodegenerative disease. While the ENS functions as an independent neural network, it also interacts with the sympathetic and parasympathetic nervous systems to maintain homeostasis [47].

### 3.1. The Brain–Gut Axis

The bidirectional relationship between the nervous system and the gut is called the brain–gut axis (BGA). The BGA is comprised of the brain (central nervous system), neuroendocrine and neuroimmune systems, the autonomic nervous system, the enteric nervous system and the gut microbiome [13,48,49]. Signaling across these pathways creates an intricate neural, hormonal and immunologic network, allowing for bidirectional modulation of the gastrointestinal and central nervous systems [49].

The hypothalamic–pituitary–adrenal axis (HPA axis) plays a key role in the brain–gut signaling pathway. The HPA axis is a neuroendocrine system responsible for maintaining homeostasis through several physiologic cascades, with one key component being regulation of the stress response. Stress stimulates the hypothalamus to release corticotropin releasing hormone (CRH), which subsequently prompts the anterior pituitary to release adrenocorticotropic hormone (ACTH). This stimulates the adrenal gland to secrete cortisol, the body’s main stress hormone. Cortisol, as well as pro-inflammatory cytokines such as IL-6 and IL-8 and neurotransmitters (i.e., norepinephrine, serotonin), have all been found to be elevated in patients with IBS [50]. Activation of these cellular effectors can lead to dysregulation of the HPA axis, which causes many of the downstream effects implicated in the pathogenesis of IBS.

The high level of psychiatric comorbidity in IBS patients is likely also related to the gut–brain axis. Although it was previously thought that IBS is a gastrointestinal manifestation of psychiatric conditions [13], data suggest that a diagnosis of IBS often precedes the onset of psychiatric symptoms in a large subset of patients [13]. The exact mechanism by which the BGA is linked to psychological disturbances is not well understood, but it is likely related to hypothalamic stress pathways, as outlined above. Aberrant metabolism of neurotransmitters may also play a role; as shown in [42,51], there is evidence that elevated levels of proinflammatory cytokines associated with IBS may increase tryptophan degradation, which may impact serotonin levels [42].

### 3.2. Factors Implicated in the Pathophysiology of IBS

Dysregulation of the BGA modifies the motor, sensory, autonomic and secretory functions of the gastrointestinal system [49], which in turn alter intestinal motility, gut permeability, visceral sensitivity and gut microbiota composition, all of which are involved in the pathogenesis of IBS [50]. Changes in intestinal motility commonly seen in IBS are thought to be mediated by altered serotonin (5-HT) metabolism. Serotonin is released from the enterochromaffin cells of the ENS to stimulate gut peristalsis and modulates secretory and vasodilator function [52]. Dysregulation of the ENS can lead to increased or reduced secretion of 5-HT, which can manifest as diarrhea or constipation, respectively [52].

There is also evidence that low-grade inflammation and immune dysfunction may play a role in IBS [13]. Patients with IBS have been found to have increased levels of pro-inflammatory cytokines, which may be partly due to stress [13,50]. Psychological stress activates an inflammatory cascade that leads to increased production of inflammatory cytokines via the HPA axis. An estimated 10% of IBS cases are post-infectious in the setting of a recent gastrointestinal illness, which often leads to mucosal and systemic inflammation [53]. Alterations in the gut microbiome also lead to several inflammatory and immunologic changes [13,54] that may compromise the gastrointestinal mucosal barrier by increasing intestinal permeability. This in turn may interfere with gastrointestinal homeostasis and dysregulate the brain–gut nociceptive pathways, leading to visceral hypersensitivity or heightened pain sensation of the gastrointestinal tract, as commonly seen in IBS [13]. For example, patients with IBS have been found to have a lower pain threshold, as demonstrated by measured responses to colonic distension [55,56].

In addition, there is evidence that dysregulation of the gut microbiota plays a role in IBS. The gut microbiota interact with the ENS and the CNS via the BGA. There is evidence that bacterial colonization promotes normal development of the ENS and CNS [57]. The gut microbiota communicate with the central nervous system in order to maintain gut homeostasis and are involved in the synthesis and regulation of neurotransmitters, immune function, maintaining the intestinal barrier, modulating the nociceptive sensory pathways implicated in visceral pain as well as intestinal permeability and gut motility [57]. Moreover, the release of neurotransmitters as part of the stress response has been linked to expression of pathogenic bacteria such as *Pseudomonas aeruginosa* and *Campylobacter jejuni* [57].

There is believed to be a genetic component to IBS, with increased risk across multiple generations of relatives [58,59]. Over 60 candidate genes have been linked to IBS, with varying levels of supporting evidence. One example is the *TNFSF15* gene, a gene that encodes the TY1A protein which is involved in the activation of the immune-cell-mediated inflammatory response in the gut mucosa [60]. Implicated genes are generally involved with serotonin metabolism, mucosal immune activation and inflammatory responses and neuropeptide signaling [60].

Diet also plays a key role in the pathogenesis of IBS, with the majority of patients able to report dietary triggers for their symptoms. Ingestion of short-chain carbohydrates or “fermentable oligosaccharides, disaccharides, monosaccharides, and polyols” (FODMAPs) has been associated with worsening IBS symptoms. FODMAPs are poorly absorbed in the small intestine, leading to increased water absorption and gas production via fermentation in the large colon [61,62], which may contribute to several of the symptoms associated with IBS, including abdominal pain and bloating [13]. The osmotic effects of ingestion of FODMAPs may also lead to distension of the gastrointestinal system and play a role in abnormal gut motility [13].

## 4. The Human Gut Microbiome

The gut microbiome comprises a myriad of intestinal microbes, including viral bacteria, fungi, and protozoa that co-exist in imparting specific functions of dietary nutrient and drug metabolism, maintenance of the gut mucosal barrier structural integrity, immunomodulation and protection against pathogens [63]. Although only an estimated one-third of the bacterial species have been identified and characterized thus far, the gastrointestinal tract is primarily comprised of *Firmicutes* (64%), *Bacteroidetes* (23%), *Proteobacteria* (8%) and *Actinobacteria* (3%) [13].

Imbalance of the gut flora may lead to a process called dysbiosis and can occur through the loss or overgrowth of a particular organism, reduction in microbial diversity or gene mutations [64]. Recent evidence suggests that gut dysbiosis may contribute to the pathogenesis of IBS. The commensal organisms that normally colonize the gut modulate signaling molecules and metabolites that are key to maintaining gut homeostasis and development of the mucosal immune system [54]. Even slight disturbances in the gut microbiome can lead to inflammatory changes that trigger oxidative stress, increase intestinal permeability and may involve bacterial translocation across the mucosal surface [54].

Key differences have been found in the gut microbiome composition in IBS patients. Although scientists have recently identified a signature gut microbiome that may be associated with severe IBS [65], characterization of the IBS intestinal microbiome remains inconsistent, and no distinct signature has been accepted [66]. In an original study of 80 patients with IBS and 65 matched controls without IBS, Jeffery and colleagues found an abundance of *Ruminococcus gnavus* and *Lachnospiraceae* and lower levels of *Barnesiella intestinihominis* and *Coprococcus catus* [67]. A meta-analysis involving 13 articles found reductions in *Bifidobacterium*, *Lactobacillus* and *Faecalibacterium prausnitzii* in IBS patients [68]. Another meta-analysis involving 16 articles and 777 patients with IBS found increased levels of Firmicutes and decreased *Bacteroidetes* (with an increased ratio *Firmicutes:Bacteroidetes* ratio) at the phylum level. They also identified several changes at lower taxonomic levels, including increased concentration of *Clostridia* and *Clostridiales* and decreased concentrations of *Bacteroidia* and *Bacteroidales* [69]. Similarly, a meta-analysis of 23 studies and 1340 subjects found lower levels of *Lactobacillus* and *Bifidobacterium,* as well as higher levels of *Escherichia coli* and *Enterobacter* in subjects in the gut microbiome analysis of patients with IBS compared to healthy controls. These researchers did not find any difference in levels of fecal *Bacteroides* or *Enterococcus* [70].

Although there is evidence that the microbiome differs between IBS patients and controls, most studies have not been able to find significant differences between IBS subtypes [67,69,71,72]. It is important to note that an inability to detect significant differences between microbiome phenotypes in people with IBS may, in part, be due to a lack of consistent methodologies. Whole shutgun metagenomics is currently the established technology used to perform analyses of gut microbial compositions; however, this approach relies on bioinformatic pipelines to interpret the data, which are associated with their own limitations [73,74]. Another point to consider is that taxonomic composition alone may not explain differences in functional phenotypes between individuals, and it is therefore important to utilize metagenomics, metatranscriptomics and metabolomics when investigating these functional differences. A recent study found that there were alterations not only in the microbiome composition of those with IBS, but also in metabolites and transcripts that relate to fructooligosaccharide utilization in those with IBS. This group also demonstrated metatranscriptomic and metabolomic differences between IBS-D and IBS-C subtypes [74].

Scientists have also identified a role that the gut virome may play in IBS, noting significantly less alpha diversity as well as different beta diversity in IBS patients, with most abundant viral clusters recognized as *Siphoviridae*, *Myoviridae* and *Podoviridae* families [72]. It is also important to note that there may be an association between the gut microbiome and psychological conditions. One study found that the fecal microbiota of patients with IBS-D was similar to that of patients with depression. Both were characterized by less overall diversity and higher abundance of *Bacteroides*, *Prevotella* or nondominant microbiota [75]. Furthermore, Peter and colleagues found that the gut microbiome in patients with IBS significantly correlated with psychological distress, anxiety and depression [76]. Further data are needed to understand the significance of this association in the development of IBS and/or psychiatric conditions. Given the wide variability in data on the microbiome in IBS, further studies incorporating metabolomic, metatranscriptomic and metagenomic sequencing are needed to better characterize the signature gut microbiome and understand its role in various diseases.

### 4.1. Small Intestinal Bacterial Overgrowth

Small intestinal bacterial overgrowth (SIBO) is another clinical condition characterized by gut dysbiosis, defined as an excess of bacteria in the small intestine. SIBO occurs secondary to fermentation of ingested carbohydrates in the gut, leading to production of gas, which manifests as abdominal pain, bloating, flatulence and altered bowel movements [77]. Diarrhea is more common than constipation and is related to immune activation, inflammation, increased intestinal permeability, enterotoxic bacterial strains and deconjugation of bile salts [77]. While SIBO is traditionally diagnosed on the basis of direct sampling and culture of jejunal aspirate, it is now more commonly evaluated by non-invasive tests, including lactulose and glucose hydrogen breath tests [78,79].

SIBO has been implicated in the pathogenesis of IBS; however, the relationship between SIBO and IBS is one that is not well delineated and remains controversial. There is considerable overlap among patients diagnosed with IBS and SIBO. One meta-analysis of 48 studies examining over 6500 patients with SIBO found that 49% of patients diagnosed by lactulose breath test and 19% of patients diagnosed by glucose breath test had a diagnosis of IBS [80]. In the literature, the prevalence of SIBO among patients diagnosed with IBS ranges from 4 to 78% and, given this variability, studies have often been scrutinized for their methodology [79]. To complicate matters, studies have found that patients with IBS are more likely to have an abnormal breath test compared to healthy controls [81,82]. Furthermore, there is evidence that patients who experienced normalization of their previously abnormal lactulose breath test after treatment with neomycin subsequently had a reduction in their IBS symptoms [81]. As previously described, it is understood that the gut microbiome is altered in IBS. There is evidence that the total number of species and bacterial colonies in the small bowel correlates with looser stool based on the Bristol stool scale [83], which may explain why patients with SIBO and IBS can experience loose stools. In addition, patients with IBS are more likely to take proton pump inhibitors due to comorbid functional dyspepsia, which may promote them to develop SIBO due to their potent antisecretory effects and hypochlorhydria [84].

It has been generally accepted that there is an association between SIBO and IBS [78,79]; however, it remains challenging to characterize the relationship. Since IBS is a clinical diagnosis based on symptoms that does not rely on ancillary testing, the exclusion of SIBO and other organic causes is not required. While it has been suggested that some patients with SIBO may have inadvertently been diagnosed with IBS due to lack of testing, this phenomenon would not account for all aspects of the pathophysiology of IBS, such as the biopsychosocial model or visceral hypersensitivity. Moreover, it would not explain why IBS responds to treatments that do not target bacterial overgrowth. High-quality studies are needed to further delineate this complex relationship.

### 4.2. Epidemiological Factors Affecting the Microbiome

#### 4.2.1. Age

The gut microbiome changes extensively across an individual’s lifespan and has been implicated in the aging process. Given the wide inter-individual variation in the microbiome, it is challenging to examine the exact role that variations in the gut microbiome play in aging. Advances in chronological age have been found to correlate increased gut biodiversity; however, overall richness of the gut microbiome has been found to decrease when using biological age with a correction for chronological age [85]. This potentially implicates age-related gut dysbiosis in individual health and longevity. Since gut dysbiosis likely triggers an inflammatory and immunological response, this may lead to age-related degeneration and epigenetic changes associated with unhealthy aging.

In a systematic review of 27 studies, Badal and colleagues found several differences in microbiota composition across different age groups, most notably increased concentration of *Akkermansia* and a relative reduction in *Faecalibacterium*, *Bacteroidaceae* and *Lachnospiraceae* [86]. Another study which assessed microbiome differences across different age groups in 153 adults grouped by age in years (<50, 50–65, 66–80, >80 years old) found that concentration of specific bacteria (*Bifidobacterium*, *Faecalibacterium*, *Bacteroides* group and *Clostridium* cluster XIVa) decreased with age in all groups up to the cohort of individuals over 80 years [87]. In adults over 80 years, the concentration of certain bacteria (i.e., *Akkermansia* and *Lactobacillus* group) increased and were associated with a significant reduction in short-chain fatty acids [87]. Interestingly, the microbiome of older adults reveals much more variation in composition than that of younger adults [88,89]. Centenarians have been found to have a particularly distinctive microbiome, with abundance of facultative anaerobes, including *pathobionts* and decreased levels of *Faecalibacterium prausnitzii* [89].

#### 4.2.2. Sex

Several studies suggest that sex may play an important role in characterizing the diversity of gut microbiota in humans. Based on the NIH Human Microbiome Project, researchers analyzed 300 individuals using 16S rRNA gene sequencing and discovered men were three times more likely to express community type D, which contains fewer *Bacteroides* and higher *Prevotella* species [90]. In population-based metagenomics analysis through the Belgian Flemish Gut Flora Project and the Dutch LifeLines-DEEP Study, sex was regarded at the 10th effect size amongst 69 variables that may significantly contribute to gut microbial variation [91]. However, the data are largely inconsistent, as several studies describe conflicting results with studies in the United States, Italy, Spain, Japan, and China [92,93,94,95,96,97]. Despite these inconsistencies, studies of gender differences in IBS reveal that IBS is more common in females compared to males (1.5- to 3-fold) and should not be overlooked as a contributor to microbial diversity in IBS [2,4].

#### 4.2.3. Ethnicity

Although not well characterized, there is evidence that ethnicity plays a role in gut microbiome composition. Studies show significant variation in regional prevalence of IBS, notably 17.5% (95% CI 16.9% to 18.2%) in Latin America, 9.6% (9.5% to 9.8%) in Asia, 7.1% (8.0% to 8.3%) in North America/Europe/Australia/New Zealand and 5.8% (5.6% to 6.0%) in the Middle East and Africa [1], which is likely in part due to genetic differences. Since diet, lifestyle, genetics and the social determinants of health are often shared by particular ethnic groups, it is challenging to understand their effect on the gut microbiome. One study which examined 16S gut microbiota data of 1673 individuals across two data sets found subtle but significant differences in the gut microbiome across different ethnic groups [98]. Another study also using 16S RNA sequencing to analyze the gut microbiome of first-generation Dutch immigrants found that individual differences in the gut microbiome could be explained, in part, by ethnic differences, including differences in α-diversity [99]. Similarly, one study using 73 white Caucasian and 182 South Asian infants from two Canadian birth cohorts found higher levels of lactic acid bacteria in South Asians and *Clostridiales* in white Caucasians [100]. Understanding the role of ethnicity in the gut microbiome may uncover why certain ethnic groups are predisposed to specific gastrointestinal conditions. For example, one study found Asian Pacific Islanders have reduced gut composition of *Odoribacteriaceae* and *Odoribacter* compared to Hispanic and Caucasian individuals, which could account for the increased severity of ulcerative colitis in Asian Americans [98]. On the other hand, another small study found similar alterations in the intestinal microbiome, notably increased abundance of *Gammaproteobacteria* and *Fusobacteria* in the microbiome of Crohn’s disease patients from Korean vs. Western populations [101]. Further studies are needed to understand how alternations in the gut microbiome due to ethnic differences may affect the development of IBS, as well as the disease severity.

#### 4.2.4. Diet

Both short- and long-term diet choices have a significant influence on the gut microbiome. A diet low in FODMAPs (fermentable oligosaccharides, disaccharides, monosaccharides and polyols) is often part of the first line of management for IBS symptoms (see below under the microbiome as a target for IBS management). Consumption of plant-based proteins is associated with an increase in commensal organisms such as *Bifidobacterium* and *Lactobacillus* and a decrease in pathogenic organisms such as *Bacteroides fragilis* and *Clostridium perfringens,* whereas consumption of animal-based protein has been associated with increased abundance of *Bacteroides*, *Alistipes* and *Bilophila* in feces [102]. On the other hand, high-fat diets have been associated with increased abundance of anaerobes and *Bacteroides* in the gut [102]. Diets high in fiber have been associated with an abundance of *Bifidobacterium* and *Lactobacillus* [103]. In addition, low-fat/high-fiber diets are associated with higher gut biodiversity [103]. Diet is a modifiable factor that modulates the gut microbiome, making it a good candidate as therapeutic intervention for conditions affected by dysbiosis.

#### 4.2.5. Pre- and Postnatal Factors

Shaping microbial colonization and diversity begins in utero [104]. Healthy colonization in early life is crucial to the development of a normal BGA [105]. Prior to birth, pre-natal factors influence gut development. Unsurprisingly, maternal diet during pregnancy has been found to impact the infant gut microbiome [106,107]. Pre-pregnancy BMI has also been associated with changes in meconium microbiome at both the genus and species level [108]. Maternal oral flora may also impact colonization of the fecal gut, as the placental microbiome is colonized by bacteria of the oral microbiome as fetuses begin to swallow amniotic fluid in the third trimester [105].

The birthing mode between vaginal delivery and cesarean section (C-section) has been speculated to influence infant microbiota [109]. One study enrolled 596 healthy full-term babies and compared the microbiota composition of the feces of both vaginal and C-section-delivered babies [110]. Vaginally delivered babies had samples enriched with *Escherichia* (*E. coli*), *Bifidobacterium* (*B. longum/breve*) and *Bacteroides/Parabacteroides species* (*B. vulgatus*, *P. distasonis*) [110], which was reinforced by other cohorts [111,112]. In contrast, C-section-delivered babies had gut microbiota resembling hospital-acquired organisms such as *Enterococcus* (E. faecalis, E. faecium), *Staphylococcus epidermis*, *Streptococcus parasanguinis*, *Klebsiella* (*K. oxytoca*, *K. pneumoniae*), *Enterobacter cloacae* and *Clostridium perfringens*, and diminished commensal bacteria [110] that resembles skin microbiota [113]. Another study showed that mode of delivery and cessation of breast-feeding were key factors in shaping infant microbiota that resembled that of the adult, demonstrating that formulation of the gut microbiome is a non-random event [112]. Although further studies are needed to understand the effects these changes in microbiome specifically have on development of IBS, preliminary data indicate that factors such as shorter duration of breastfeeding, C-section birth or low birth weight are associated with increased development of IBS in adulthood [114,115].

### 4.3. The Microbiome as a Target of IBS Management

#### 4.3.1. Diet

The majority (60%) of patients with IBS report dietary triggers to their gastrointestinal symptoms [116]. A diet low in FODMAPs (fermentable oligosaccharides, disaccharides, monosaccharides and polyls) is often recommended as part of management of IBS. These small-chain carbohydrates (FODMAPs) have poor absorption in the small intestine, increasing intestinal osmolality, leading to increased water absorption and gas production via fermentation in the large colon [61,62]. Thus, eliminating them from one’s diet has been shown to reduce symptoms of IBS.

One meta-analysis of 22 studies (6 RCTs and 16 non-randomized trials) found that subjects who adhered to a low-FODMAP diet had lower overall symptom severity scores, in addition to improvement in quality of life and abdominal pain [117]. Another meta-analysis consisting of data from 9 RCTs found that a diet low in FODMAPs yielded significant improvement in GI symptoms, abdominal pain and health-related quality of life compared to other diets [116]. Given the challenges associated with adhering to a low-FODMAP diet, recent studies have compared the low-FODMAP diet to other modified but less restrictive diets for IBS symptom relief. One small-scale RCT that compared the low-FODMAP diet to a low lactose diet in 29 IBS patients found that both diets significantly reduced the IBS Severity Scoring System scores to a similar extent; however, the low-FODMAP diet was more effective at reducing abdominal pain and bloating than the low-lactose diet [118]. Another RCT which compared the low-FODMAP diet to the modified NICE diet (a diet modified from the National Institute of Health and Care Excellence guidelines that recommends eating small frequent meals, avoiding known triggers and avoiding alcohol and caffeine intake) in IBS-D patients found that both diets were associated with improvement in symptoms; however, the low-FODMAP diet was more effective at ameliorating abdominal pain and bloating compared to the mNICE diet [119]. Similarly, a meta-analysis of 10 studies comparing the FODMAP diet to a traditional IBS diet (i.e., high fiber, low fat) found that although both can be effective, IBS-SSS scores were significantly lower in the FODMAP diet group [120]. On the other hand, a small study of 28 subjects who trialed three diets (low-FODMAP, gluten-free and balanced diets) for 4 weeks each found that all three diets reduced symptoms of abdominal pain and bloating, symptom severity and improved quality of life; however, 86% of subjects preferred the balanced diet [121].

It has been suggested that the gut microbiome could be used as a biomarker to recognize which IBS patients may experience the most benefit from the low-FODMAP diet. One study examining the effects of the low-FODMAP diet in children with IBS found that participants who experienced improvement of symptoms with the FODMAP diet had higher levels of specific taxa (*Bacteroides*, *Ruminococcaceae*, *Faecalibacterium prausnitzii*) that are associated with increased rates of saccharolytic metabolism [122]. Unfortunately, the influence that the low-FODMAP diet has on the microbiome regarding symptomatic relief remains elusive. Studies have found a reduction in luminal *Bifidobacteria* after a low-FODMAP diet [116], which is counterintuitive, as luminal *Bifidobacteria* has been found to be lower in IBS patients compared to healthy people, and administration of pro- and pre-biotics containing high concentrations of *Bifidobacteria* has been associated with improved symptoms [123]. Despite substantial evidence to support that a diet low in FODMAPs may improve symptoms in IBS patients, further investigation is warranted to observe the long-term effects of modulating the diet on the composition of the microbiome, and the effects of those changes on IBS symptoms.

#### 4.3.2. Pre- and Probiotics

Prebiotics are indigestible compounds consisting of carbohydrates (fructooligosaccharides [FOS] and galactooligosaccharides [GOS]) which stimulate the growth of bacteria [123]. As described above, lower levels of *Bifidobacteria* and *Lactobacilli* have been noted in the microbiome of IBS patients compared to healthy controls. Since levels of these bacteria have been found to increase with consumption of FOS and GOS [123,124], consumption of prebiotics has been suggested as a possible treatment for IBS. In addition to modulation of the microbiome, it has been suggested the anti-inflammatory and antioxidative effects of prebiotics may be beneficial for IBS symptoms [125]. Prebiotics also alter stool consistency by increasing stool bulk and fecal water content, which may be beneficial in constipation-predominant IBS [66,125].

Data investigating the role of prebiotics as a potential treatment for IBS are sparse and contradictory. Several studies have found no significant change in IBS-associated symptoms with administration of prebiotics vs. a placebo [126,127]. In a parallel crossover-controlled study of 44 participants with IBS, Silk and colleagues found that patients who received either 3.5 g/d or 7 g/d trans-GOS prebiotic for 12 weeks had significantly increased levels of *Bifidobacteria* in their stool compared to the placebo group [128]. Notably, participants who received the low-dose prebiotic (3.5 g/d) reported significant changes in stool consistency, decreased flatulence, decreased composite score of symptoms (abdominal pain, bloating, ease of bowel movements) and improvement in subjective global assessment (SGA) scores [128]. Those who received the high-dose prebiotic (7 g/d) had improved SGA and anxiety scores [128].

Probiotics are live microorganisms that have beneficial properties specific to the gut microbiota when consumed. A recent meta-analysis of 35 RCTs found that consumption of probiotic resulted in a significant improvement in symptoms of IBS, including abdominal pain, bloating, and flatulence [129], results that are consistent with findings of several other meta-analyses [130,131,132,133]. The impact of the gut on brain signaling has been associated with the consumption of probiotics.

Probiotics also appear to exert an influence on mood via the BGA. For instance, a randomized, double-blind, placebo-controlled study of adults with IBS treated with the probiotic *Bifidobacterium longum* NCC3001 resulted in a reduction in depression scores and decreased responses in the amygdala and fronto-limbic regions seen on fMRI when exposed to negative emotional stimuli [134]. However, anxiety and quality of life with IBS were unchanged in the probiotic-treated group [134]. Although there are no current established guidelines on the use of probiotics in treatment of depression, a meta-analysis demonstrated that probiotics in supplementation to antidepressants showed a significant positive effect of probiotics on depressive symptoms [135].

In addition to modulation of the microbiome composition, several other theories have been proposed regarding the benefits associated with probiotic consumption for IBS symptoms. It has been suggested that probiotics improve mucosa barrier function and reduce intestinal permeability, which has been implicated in the pathogenesis of IBS [132,133]. There is also evidence that probiotics induce production of cytokines including IL-10, modulating the host immune response [136].

Despite evidence across many RCTs that probiotics have beneficial effects in symptom management in IBS patients, there is considerable variability in findings across studies that warrant further clarification to guide treatment. In a meta-analysis of 37 trials and 4403 subjects, Ford and colleagues found a statistically significant symptomatic improvement associated with use of prebiotics; however, noted significant heterogeneity as well as notable publication bias or other small study effects challenge the overall validity of the data [137]. To cite some specific discrepancies among data sets, Asha et al. found that probiotics containing lactobacillus were associated with improvement in abdominal pain, flatulence scores and quality of life, whereas Bifidobacterium improved urgency and global IBS symptoms [132]. Thus, they postulated that multi-strain probiotics could offer better symptomatic improvement in IBS. On the other hand, Zhang and colleagues found that single-strain probiotics were more effective in regard to overall symptom response compared to multi-strain probiotics [130]. Zhang and colleagues also found that shorter durations of treatment (i.e., less than 8 weeks) may be more effective in terms of global symptoms and quality [130].

Given the limited data available, and the significant heterogeneity of existing studies, the use of prebiotics and probiotics for treatment of IBS has not been generally accepted by the Gastroenterology community. At present, the American Gastroenterological Association has no official recommendations regarding the use of probiotics for treatment of IBS [138]. In fact, the American College of Gastroenterology guidelines recommend against probiotics for treatment of IBS, acknowledging that their recommendation is conditional based on a very low level of evidence [139]. Additional large-scale RCTs are needed to further clarify the role of pre- and probiotics in the treatment of IBS.

#### 4.3.3. Antibiotics

Given strong evidence that dysbiosis may play a role in the pathogenesis of IBS, antibiotics have been targeted as a potential treatment for IBS. The antibiotic commonly implicated in the treatment of IBS is Rifaximin. Rifaximin is a broad-spectrum oral antibiotic with negligible systemic absorption, a favorable side effect profile and low evidence of resistance [140]. Two double-blinded multi-center randomized controlled trials—TARGET 1 and TARGET 2—administered 550 mg Rifaximin vs. placebo to patients with IBS-D three times daily for 2 weeks and found that patients in the Rifaximin group vs. the control group showed overall improvement in IBS symptoms (40.7% vs. 31.7%, *p* < 0.001) one month post-treatment [141]. A meta-analysis examining data from five randomized controlled trials, including the TARGET data comparing Rifaximin to placebo in the treatment of IBS, found that administration of the antibiotic yielded a statistically significant improvement in IBS symptoms (RR 0.84; 95% CI 0.79–0.90) [137]. Data comparing administration of Rifaximin to other antibiotics including neomycin, doxycycline, amoxicillin/clavulanate and ciprofloxacin have found Rifaximin to be more efficacious with lower concern for resistance [142]. Per the TARGET 1 and 2 trials, treatment of IBS with Rifaximin also has a favorable side effect profile without associated cases of *Clostridium difficile* [141]. At present, the *American Journal of Gastroenterology* formally recommends the use of Rifaximin for treatment if IBS-D [139].

The precise mechanism by which antibiotics alleviates IBS symptoms is not well understood. It has been postulated that antibiotics alter the gut microbial composition, reducing harmful bacterial products and altering inflammatory and immune responses [141]. There is limited data to support that reduced hydrogen excretion on a lactulose breath test after administration of antibiotics is associated with a reduction in IBS symptoms [81,143]. Studies have found a change in the gut microbiota in patients with IBS post-treatment with Rifaximin. One study which found that fecal samples from subjects with IBS had significantly greater species richness (number of species per sample) compared to healthy controls demonstrated a significant reduction in fecal richness after treatment with Rifaximin, despite a stable Bacteroidetes/Firmicutes ratio [144]. Similarly, another study found a lower relative abundance of select bacteria directly after treatment with Rifaximin in IBS patients; however, these changes were not sustained at the end of the study period, suggesting the changes in gut composition may be transient [145]. Further data will need to be collected to further delineate the effect of Rifaximin on the microbiome.

#### 4.3.4. Fecal Microbiota Transplantation

Fecal microbiota transplantation (FMT) is the process in which stool from a healthy donor is transferred to the colon of a different individual with the intention of altering their gut microbiome. FMT has been studied as a safe and efficacious treatment for Clostridium difficile infection [123]; however, only recently has it been examined as a potential treatment for IBS, as well as several other gastrointestinal conditions.

Studies examining the efficacy of FMT in patients with IBS have found conflicting results (Table 1). In a double-blinded placebo-controlled RCT of 165 patients who received 30 g FMT, 60 g FMT vs. placebo (own feces) via gastroscope at a 1:1:1 ratio, El-Salhy et al. found significant improvement of IBS symptoms 3 months post-FMT in both experimental groups compared to placebo [146]. In another double-blinded placebo-controlled RCT, Johnsen and colleagues assigned 90 patients (2:1) to receive FMT via colonoscope vs. placebo (own feces), finding a significant improvement in IBS symptoms based on IBS-SSS scores post-FMT in the experimental group [147]. Similarly, in another double-blinded RCT, Holvoet and colleagues found that patients with refractory IBS who underwent a nasojejunal administration of donor stool experienced a significant improvement in IBS-related symptoms compared to patients in the placebo group who received autologous stool [148]. On the other hand, in a double-blinded placebo-controlled RCT of 52 participants who received FMT vs. placebo capsules for 12 days, Halkjær and colleagues found a significant reduction in the IBS-severity scoring system scores (IBS-SSS) and quality of life scores after 3 months in patients who received the placebo [149]. In a double-blinded RCT, Madsen and colleagues found that FMT administered in capsule form for 12 days did not significantly improve abdominal pain, stool frequency or stool form in patients with moderate-to-severe IBS during treatment or at one, three or six-month follow up; however, they did note a statistically significant improvement in stool frequency in the FMT group when examining improvement in stool frequency during treatment to post-treatment and at one month [150]. Similarly, a meta-analysis which pooled data from 254 participants across four studies found no significant improvement in IBS symptoms in patients who received FMT versus placebo at 12 weeks [151]. Another meta-analysis which contained data from 5 RCTs and 267 patients found that IBS symptoms did not significantly improve post-FMT regardless of stool type [152]. Given a substantial amount of conflicting data, further investigation is needed to characterize the possible role of FMT in treatment of IBS.

To better understand the effect of FMT on treatment of IBS, studies have also collected data on changes to the gut microbiome post-FMT. El-Salhy and colleagues found an increased concentration of *Eubacterium biforme*, *Lactobacillus* spp. and *Alistipes* spp. and a reduced concentration for *Bacteroides* 1 month post-FMT in both experimental arms [146]. Furthermore, the concentration of *Alistipes* spp. and *Lactobacillus* spp. correlated negatively with the IBS-SSS score, suggesting the change in gut composition may be clinically significant [146]. The authors also found a decrease in the presence of dysbiosis in the 60 g FMT group from 61% to 39% post-FMT; however, the finding was not statistically significant (*p* = 0.108) [146]. Halkjær and colleagues found that the microbiome of fecal donors was significantly more diverse than the microbiome of the patients with IBS [149]. Moreover, the microbiome of patients with IBS who received the FMT capsules was not significantly different from the donors’ microbiome post-FMT [149]. Interestingly, alpha-diversity in patients with IBS (in the experimental and control arm) did not correlate with the IBS-SSS scores [149]. Further data are needed to clinically correlate changes in gut microbiome composition with symptomatic relief in IBS patients, as this may lead to more efficacious treatment.

#### 4.3.5. The Interaction between Antidepressants, the Gut Microbiome and IBS

Recent meta-analyses have found that antidepressants—particularly tricyclics and selective serotonin reuptake inhibitors—improve global IBS symptoms as well as abdominal pain [153]. While SSRIs demonstrate a statistically significant benefit for improvement of global symptoms, TCAs appear to have an advantage in improvement of abdominal pain and symptom score [154]. The mechanisms by which antidepressants improve symptoms remain unclear. Antidepressants are purported to interfere with afferent signals from the gut to the CNS, thus improving abdominal pain. Other mechanisms that have been cited, specifically with Citalopram, include decreased sensitivity of the colon to distension, accelerated transit time [155] and an affective memory bias towards positive material, reducing attention to gastrointestinal sensations [156]. Psychotropics also influence the HPA axis and modulate efferent sympathetic and parasympathetic efferent signals to the gut, which may also improve IBS symptoms [157].

In recent years, there have been more studies investigating the relationship between the effect of antidepressants on IBS and the gut microbiome. Studies have found changes in the composition, diversity and abundance of virulence factors in the gut microbiome in patients with IBS compared to controls [158]. As mentioned previously, there is no widely accepted “signature” microbiome differentiating IBS patients from healthy controls, although there are data to support a distinct intestinal microbial profile differentiating severe IBS from mild/moderate [65]. Similarly, it has been shown that patients with major depressive disorder have a different gut microbial landscape than healthy controls [159]. Jiang et al. found that even when accounting for interindividual variability, several predominant genera, including the *Enterobacteriaceae*, *Alistipes* and *Faecalibacterium*, were present in different quantities in depressed individuals compared to healthy controls [159]. Their study also found a negative correlation between *Facalibacterium* and the severity of depressive symptoms [159].

The degree to which microbial changes in IBS patients correlate with changes reported in depressed patients has not been well studied. There are also few studies to date examining whether antidepressants exert some of their clinical effects via modification of the gut microbiome. In one study in mice, administration of a bacterial strain of *R. flavefaciens* was found to attenuate the antidepressant effect of duloxetine [160]. The specific outcome these authors measured was immobility as a marker of depression in mice. Their findings suggest that different microbiota can have a direct influence on the efficacy of antidepressants. Interestingly, this finding parallels the report of synergistic effects of probiotics and antidepressants on depressive symptoms [160].

It is well established that the use of antidepressants is associated with changes in the microbiome [159,161]. Chait et al. found different SSRIs have antimicrobial effects to varying degrees, and these affect different species more than others. Specifically, they studied six different antidepressants with varying mechanisms of action and found desipramine and aripiprazole to have the most antimicrobial activity [161]. They also found that the bacterial strains *Akkermansia muciniphila*, *Bifidobacterium animalis* and *Bacteroides fragilis* were most vulnerable to antimicrobial activity [161]. The antimicrobial action of SSRIs was also supported by McGovern et al., who postulated that SSRIs exert their influence on the gut microbiota via inhibition of efflux pumps and/or amino acid transporters [162]. Future studies in humans are required to establish the directionality of the relationship between antidepressants and the gut microbiome specifically on IBS outcomes.

## 5. Discussion

Irritable bowel syndrome is a highly prevalent functional gastrointestinal disorder characterized by chronic abdominal pain or discomfort and altered bowel habits. IBS has a high disease burden, frequently manifesting with debilitating extra-intestinal comorbidities, including anxiety and depression, often leading to increased healthcare utilization and costs, decreased work productivity and reduced quality of life. It has been postulated that disturbances in the brain–gut axis, as well as dysbiosis, may be implicated in the pathophysiology of IBS, with the gut microbiome driving some of these changes. There are data to suggest that certain signature changes in the gut microbiome are associated with IBS; however, no IBS-specific microbiome has been clearly identified. Moreover, treatments that target the microbiome in IBS patients, including pre- and pro-biotics and fecal microbiota transplants, have failed to show consistent results or sustained improvement in symptoms. Further investigation is warranted to investigate the role of precision medicine and metabolomics in targeting the microbiome as a treatment of IBS on an individual basis.

## Figures and Tables

**Figure 1 jcm-12-02558-f001:**
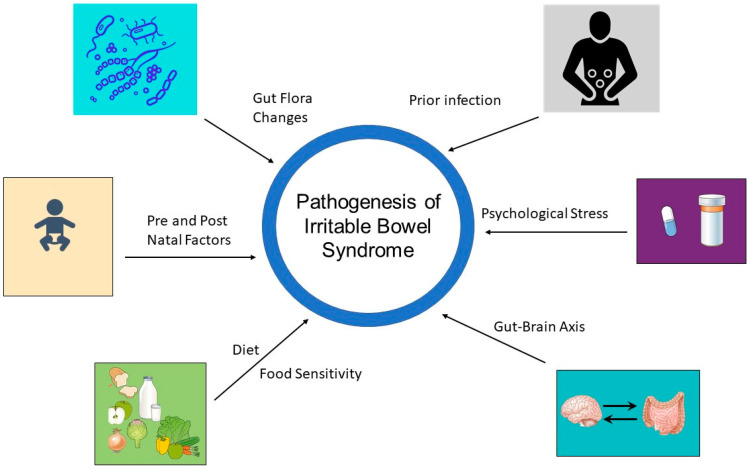
Elements in the Pathophysiology of Irritable Bowel Syndrome.

**Table 1 jcm-12-02558-t001:** Selected fecal microbiota transplantation studies.

Author	Study Design	Methods	FMT Administration	Results
El-Salhy [146]	Single-center, double-blind, placebo-controlled RCT	165 patients: placebo (*n* = 55), 30 g FMT (*n* = 54), 60 g FMT (*n* = 55)	Gastroscope to the distal duodenum	Clinical response * 23.6%, 76.9% and 89.1% for placebo, 30 g, and 60 g
Johnsen [147]	Single-center, double-blind, placebo-controlled RCT	83 patients: placebo (*n* = 28) and 50–80 g FMT(*n* = 55)	Colonoscope to the cecum	Clinical response ** 65% (FMT) vs. 43% (placebo)
Holvoet [148]	Single-center, double-blind, placebo-controlled RCT	62 patients: placebo (*n* = 19) and FMT (*n* = 43)	Nasojejunal probe	Clinical response *** 56% (FMT) vs. 26% (placebo)
Halkjær [149]	Single-center, double-blind, placebo-controlled RCT	45 patients: placebo (*n* = 23) and FMT(*n* = 22)	Capsule form	Clinical response ** 79% (placebo) vs. 36% (FMT)
Madsen [150]	Single-center, double-blind, placebo-controlled RCT	51 patients: placebo (*n* = 26) and FMT (*n* = 25)	Capsule form	No difference in clinical response **** during treatment or one, three or six months post-treatment

Studies examining the efficacy of FMT in patients with IBS are not standardized, relying on different dosage and administration methods of FMT, and overall have had conflicting results. While El-Salhy et al. and Johnsen et al. found a favorable clinical response with FMT vs. placebo, Halkjær and colleagues found a significant improvement in IBS symptoms in the placebo group compared to FMT. A complete compilation of registered clinical trials with the NIH is provided in the Appendix A (NIH website clinicaltrials.gov, accessed on 9 March 2023). Abbreviations: RCT (randomized control trial); FMT (Fecal microbiota transplantation). * defined by IBS symptom score. ** defined by IBS severity score. *** defined by a daily symptom diary. **** defined by abdominal pain and stool frequency (daily symptom diary) and stool form (weighted stool score).

## Data Availability

Not applicable.

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
