# Peer review of "Irritable Bowel Syndrome and the Gut Microbiome: A Comprehensive Review"

_jcm, 2023, doi:10.3390/jcm12072558_

Round 1
Reviewer 1 Report
Review report Journal of Clinical Medicine
Irritable Bowel Syndrome and the Gut Microbiome: A Compre-2 hensive Review
By
Sofia D Shaikh 1, Natalie Sun 1, Andrew Canakis 2, William Y Park 3, H Christian Weber
This is a well written and nicely stuctured review on a quite complex topic. I think the described aspects of the topic are well presented, with relevant references. A few supplemental considerations could be added – please see below.
One important point in discussion of microbiome data: there is no general consensus on sample processing, technology, in-silico pipelines as well as which reference databases to use. In our metagenome dataset we got completely different results from the same raw data using KEGG compared to Kaiju. This may explain some of the discrepancies between «IBS defining taxa profiles» in different studies - even the feature of reduced microbiome diversity in IBS can be debated. Hence, some of the great variability in reported profiles may in part be due to differences in methodology.
Another point is that taxonomy alone is a poor explanation of differences between microbiomes. The functional phenotype of same taxa may differ quite a lot depending on gene pool differences like plasmids or cross species interactions like quorum sensing as well as availability of different environmental substrates that may affect gene utilization. Functional metagenomics and metabolomics may be more important in explaining the functional features of the microbial society as a whole.
Fig 1: Figure is low resolution and should be rendered in high quality. I find it more logical to reverse the arrows indicating that the different components affect the pathogenesis of IBS and not the other way round. The presented figure may imply that an exclusive reason exist for each patient with IBS, while probably a combination of different factors lead to the Gi disturbance.
Section 3: A bit diffuse on the description of the gut-brain axis. Better in section 4.1. May want to restructure?
Line 202 missing word? Sentence ending a bit abruptly
Supplemental on BGA: «false» neurotransmitters may be a pathway for microbiome to affect brain function. Most of the serotonin in the brain is derived form tryptophane of microbial origin.
Author Response
Review report Journal of Clinical Medicine
Irritable Bowel Syndrome and the Gut Microbiome: A Compre-2 hensive Review
By
Sofia D Shaikh 1, Natalie Sun 1, Andrew Canakis 2, William Y Park 3, H Christian Weber
This is a well written and nicely stuctured review on a quite complex topic. I think the described aspects of the topic are well presented, with relevant references. A few supplemental considerations could be added – please see below.
One important point in discussion of microbiome data: there is no general consensus on sample processing, technology, in-silico pipelines as well as which reference databases to use. In our metagenome dataset we got completely different results from the same raw data using KEGG compared to Kaiju. This may explain some of the discrepancies between «IBS defining taxa profiles» in different studies - even the feature of reduced microbiome diversity in IBS can be debated. Hence, some of the great variability in reported profiles may in part be due to differences in methodology.
Another point is that taxonomy alone is a poor explanation of differences between microbiomes. The functional phenotype of same taxa may differ quite a lot depending on gene pool differences like plasmids or cross species interactions like quorum sensing as well as availability of different environmental substrates that may affect gene utilization. Functional metagenomics and metabolomics may be more important in explaining the functional features of the microbial society as a whole.
Our Response: Thank you for these critical, thoughtful points. We have added a discussion in the microbiome section (Section 4) addressing these limitations. See lines 316-329.
Fig 1: Figure is low resolution and should be rendered in high quality. I find it more logical to reverse the arrows indicating that the different components affect the pathogenesis of IBS and not the other way round. The presented figure may imply that an exclusive reason exist for each patient with IBS, while probably a combination of different factors lead to the Gi disturbance.
Our Response: The reviewer raises a good point: this figure has been recreated with reverse of the arrows and has been added back to the manuscript in higher resolution.
Section 3: A bit diffuse on the description of the gut-brain axis. Better in section 4.1. May want to restructure?
Our Response: We have moved the brain-gut axis (BGA) section to section 3.1 (lines 188 ff.) instead of section 4 to minimize redundancy and improve the structure of the manuscript.
Line 202 missing word? Sentence ending a bit abruptly
Our Response: This sentence was cut off and has been updated within the newly structured section 3 (Pathophysiology of IBS) starting line170 (lines 170 -272).
Supplemental on BGA: «false» neurotransmitters may be a pathway for microbiome to affect brain function. Most of the serotonin in the brain is derived form tryptophane of microbial origin.
Our Response: Thank you for pointing out this error. We have decided to remove the figure and supplemental text from the article.

Reviewer 2 Report
Shaikh et al provide a comprehensive review on IBS and data surrounding the potential role of the microbiome in driving IBS symptoms. The review is sufficiently detailed, however, there are some deficiencies.
Major comments:
1. The authors do not cover Small Intestinal Bacterial Overgrowth and its treatment with antibiotics, which relieve IBS symptoms. I suggest that a section covering this topic is included as it fits the remit of the review.
2. The Enteric Nervous System (ENS) and autonomic nervous are well known mediators of gastrointestinal function, where dysfunction of these pathways can mediate sensitisation associated with pain, the key symptom which determines the classification of IBS according to Rome IV criteria. The authors need to discuss these neuronal systems in the review describing both the roles they play in symptom generation (including motility dysfunction leading to constipation/diarrhoea) and relationship with the gut microbiome e.g. effects of gut bacteria on ENS neuroanatomy/function.
3. Figure 2 is not demonstrative of information that is useful nor does it provide any detail relating to interacting factors between the Gut-Microbiome-Brain. The figure should either be removed or edited so it has detail, i.e. providing a summary of environmental, microbial, dietary, psychosocial factors that collectively drive IBS symptom generation.
Minor comments:
1. In section 4.3.2, the title should be changed to "Pre- and Probiotics" instead of "Pre and Pro Biotics" as currently stated.
Author Response
Shaikh et al provide a comprehensive review on IBS and data surrounding the potential role of the microbiome in driving IBS symptoms. The review is sufficiently detailed, however, there are some deficiencies.
Major comments:
- The authors do not cover Small Intestinal Bacterial Overgrowth and its treatment with antibiotics, which relieve IBS symptoms. I suggest that a section covering this topic is included as it fits the remit of the review.
Our Response: Thank you for this excellent suggestion. We have added a section on the discussion of SIBO (Section 4.1; lines 347 ff, 347-388).
- The Enteric Nervous System (ENS) and autonomic nervous are well known mediators of gastrointestinal function, where dysfunction of these pathways can mediate sensitisation associated with pain, the key symptom which determines the classification of IBS according to Rome IV criteria. The authors need to discuss these neuronal systems in the review describing both the roles they play in symptom generation (including motility dysfunction leading to constipation/diarrhoea) and relationship with the gut microbiome e.g. effects of gut bacteria on ENS neuroanatomy/function.
Our Response: Thank you for this insightful suggestion. While we did discuss several ways that dysregulation of the ENS and ANS is implicated in the pathogenesis of IBS, we did not explicitly discuss the ENS. We have now specifically discussed the ENS in Section 3 (lines 170-187; section 3.1., lines 190-195; section 3., lines 219 ff): The Pathophysiology of IBS. We have also expanded on the role that the ENS and ANS play in the pathogenesis of IBS when discussing gut motility, inflammatory and immune dysregulation, visceral hypersensitivity, and the gut microbiome.
- Figure 2 is not demonstrative of information that is useful nor does it provide any detail relating to interacting factors between the Gut-Microbiome-Brain. The figure should either be removed or edited so it has detail, i.e. providing a summary of environmental, microbial, dietary, psychosocial factors that collectively drive IBS symptom generation.
Our Response: We agree and have decided to remove this figure.
Minor comments:
- In section 4.3.2, the title should be changed to "Pre- and Probiotics" instead of "Pre and Pro Biotics" as currently stated.
Our Response: This has been updated in the review article.

Round 2
Reviewer 2 Report
Accept for publication